# Entrepreneurship Experiences among Vietnamese Marriage Immigrant Women in Taiwan

Ya-Ling Wu

Graduate Institute of Technological and Vocational Education, National Pingtung University of Science & Technology, Pingtung 912, Taiwan; karin@mail.npust.edu.tw

**Abstract:** Since the 1990s, Taiwan has experienced growing numbers of commercially arranged marriages between Vietnamese women and socioeconomically disadvantaged Taiwanese men. Most Vietnamese marriage immigrant women proactively engage in the labor market due to the heavy financial burden of their Taiwanese and natal families. Employing a sociocultural and post-structural feminist approach, this study draws from life-story interviews of 13 married Vietnamese women to investigate the entrepreneurship experiences among Vietnamese marriage immigrant women in Taiwan. These women are pushed and pulled towards creating demanding micro-entrepreneurships based on their self-employed socialization, thereby fulfilling family obligations and achieving career goals. Targeting their host market, these women operate their businesses using Taiwanese customer networks and their institutionalized learning and sustainable resilience while negotiating self-identity. Running entrepreneurships empowers these women, facilitating their self-identity, social integration, family position within the boundaries of gender, family expectations, and business while they struggle with unexpected challenges. Clearly, these individuals and their significant others, homeland culture and socialization, and their life experiences and positions in Taiwan shape these immigrant women's businesses and their sense of meaning. This study extends the feminist perspective of this topic, focusing on the sustainable agency and sense of competence of female marriage immigrants.

**Keywords:** immigrant women; entrepreneurship experiences; sociocultural approach; post-structural feminist theories; marriage migration; sustainable agency

## 1. Introduction

Over the last three decades, a rapidly increasing number of women in less economically developed regions of southeastern Asia have married men in wealthy areas of eastern Asia to obtain a better life. These women constitute one sector of the essential international migration flow [1]. Since the 1990s, Taiwan has experienced growing numbers of commercially arranged marriages between Vietnamese women and socioeconomically disadvantaged Taiwanese men. Prior to 2021, more than 108,000 Vietnamese women migrated to Taiwan through transnational marriage [2], and therefore comprise a new vulnerable population in Taiwan due to their complicated statuses, which involve gender, marriage migration, and class [3]. In Taiwan, Vietnamese immigrant women are usually expected to act as a subservient wife and a responsible mother. Many Taiwanese husbands are more likely to restrict their immigrant wives within narrow social networks and grant them limited access to information in order to keep them from running away. Additionally, these immigrant women tend to contribute to their natal families' finances. Therefore, some Taiwanese usually stigmatize Vietnamese marriage immigrant women and believe that immigrant women come to Taiwan only to make money [4].

Furthermore, Vietnamese married immigrant women seem to play instrumental roles in the Taiwanese family by continuing their husband's family line through producing the next generation and supporting the household through outside employment [4]. In fact, these women disproportionately work as secondary and necessary laborers in low-paid

menial jobs, thereby contributing to the Taiwanese economy. Most Vietnamese female marriage immigrants proactively engage in the labor market due to the heavy financial burden of their Taiwanese and natal families. In 2018, in Taiwan, the labor force participation rate of immigrant women (67.9%) exceeded that of their Taiwanese counterparts (58.2%). For these working immigrant women, the self-employment (i.e., working as employers or own-account workers) rate was 18.3%, clearly outweighing that of native-born women (8.40%). Among these self-employed immigrant women, 40.4% worked in the beauty industry, 23.0% in the catering industry, 14.1% in the personal service industry (e.g., personal caring and cleaning), and 5.2% in the retail industry. Most female immigrant entrepreneurs run a micro-business, with employees ranging from zero to five, significantly contributing to the Taiwanese economy [5,6]. However, their entrepreneurial activities have attracted little scholarly attention.

Although some studies have explored female immigrant entrepreneurship, most of them only highlight this issue in developed areas, such as North America, Western Europe, and Austria; involve well-established ethnic groups, such as Chinese people, Koreans, Indians, and Latinos; and emphasize immigrant women who migrate under family reunification [7,8]. However, few studies have concentrated on the self-employment experiences of female Asians who move to other Asian areas through commercially arranged marriages. This neglect may originate from the fact that, in patriarchal host societies, marriage immigrant women are viewed as wives and mothers rather than as workers [9]. Hence, these factors have collectively created a gap in the comprehensive knowledge of the contemporary entrepreneurship experiences of Vietnamese marriage immigrant women in Taiwan.

Previous scholars initially emphasized the effect of either individual characteristics or ethnic culture on immigrant entrepreneurship. Furthermore, an interaction model was utilized to examine the interaction between the group characteristics of different immigrants and the determinants for operating their businesses [7]. By focusing on the multiple contexts in which immigrant business functions, the mixed embeddedness approach developed by Koosterman and Rath [10] has investigated immigrant entrepreneurship more deeply through the complex framework of opportunity structures in host countries. Nevertheless, this perspective overlooks immigrants' early socialization in their country of origin, which may affect their self-employment [11]. Additionally, these four approaches, rooted in Western countries, have been particularly applied to examine male immigrant entrepreneurship. Even within the literature on female immigrant entrepreneurship, most studies have applied an androcentric perspective without highlighting any gendered dimensions [8].

To ensure a comprehensive understanding of female immigrant entrepreneurship, it is necessary to adopt a more dynamic approach through a feminist perspective that thoroughly considers the interaction between female immigrant entrepreneurs and the complicated contexts within their country of origin and the country they move to. Thus, by employing a sociocultural and post-structural feminist approach, this study aims to investigate the entrepreneurial experiences of Vietnamese marriage immigrant women in Taiwan and the meanings that they construct regarding these experiences. The key questions of this research are as follows: What are Vietnamese marriage immigrant women's motivations for starting their own business in Taiwan? How do these women operate their entrepreneurship in the Taiwanese labor market? How do their prior socialization in Vietnam and their sociocultural contexts in Taiwan affect immigrant women's entrepreneurship in Taiwan? How does undertaking entrepreneurship in Taiwan affect these female immigrants?

This study makes some specific academic contributions. First, it explores Asia–Asia female migration, an essential but largely uninvestigated sector of international migration dynamics, to present a new context for female immigrant entrepreneurship. Second, this study emphasizes Vietnamese marriage immigrant women, a previously relatively unknown group. Third, it highlights how a sociocultural approach and feminist theories help to understand, and are enriched by, discussion on this issue.

## 2. The Study Framework

### 2.1. The Sociocultural Approach

Sociocultural theories are grounded in Vygotsky's work and suggest that individual thoughts and activities are a personal process that is shaped by and shapes the interaction between individuals and the multiple contexts of social, cultural, institutional, physical, and historical settings within which they are living and have lived [12]. Cultural tools, such as identity and voice, which people use in daily activities, mediate their experiences [13]. This framework assumes that all individuals, as members of multiple defined communities, participate in practices through their diverse cultures and experience of both stability and change and homogeneity and heterogeneity. Furthermore, individual experiences must be understood within and across spaces [14]. Thus, immigrant women construct their entrepreneurial experiences based on what they already know, and all the tools they employ integrate into their entrepreneurship, which can be situated, social, and dynamic [15].

### 2.2. Poststructural Feminism

By analyzing power and authority via the interrelations of subjectivity, language, discourse, social institutions, and agency at the intersections of gender, class, ethnicity, and other categories, post-structural feminism aims to facilitate social equity by challenging the status quo [16]. This framework assumes that women's self-identity, which is comprised of their positionalities, is relative to and constantly shifting amid changing contexts [17]. Because positionality and voice are considered means of constructing reality, this approach conceptualizes the possibility of individual sustainable agency, even in the face of the most severe challenges [18]. The sustainable agency that post-structural feminism opens up is a recognition of the power to change and a fascination with the capacity to generate new life-forms [19].

In sum, based on the sociocultural approach and post-structural feminism, which intersect through the gendered and sociocultural contexts in which women live, this study analyzes the entrepreneurship experiences among Vietnamese marriage immigrant women in Taiwan by connecting their personal entrepreneurships with sociocultural contexts and emphasizing their self-identity and sustainable agency.

## 3. Methodology

A qualitative interpretive approach was employed to understand the complexity and meanings of entrepreneurial activities among female Vietnamese marriage immigrants in Taiwan [20].

### 3.1. Participants

The participants were 13 Vietnamese immigrant women who entered Taiwan through commercially arranged marriages with Taiwanese men. The snowball method was used to recruit the participants [21], with their informed consent, according to their business sector and duration in Taiwan, to generate typical cases of Vietnamese female immigrant entrepreneurship. At the time of interviews, each participant had worked as an entrepreneur; namely, each had been self-employed as an own-account worker or employer for at least 3 years in Taiwan, providing sufficient entrepreneurship experience to reflect on. The participants' average age was 38.3 years, and their average duration in Taiwan was 14.8 years. Most had obtained a Vietnamese high school education. All except one had at least one child. Four participants were divorced, and two were widowed. All women started their micro-business by running shops or being vendors in the catering, beauty, sales, agriculture, or tailoring sectors. Table 1 presents the participants' backgrounds.

**Table 1.** The background of the participants.

| Name Age | Marital Status Number of Children | Education in Vietnam | Length of Residence in Taiwan | Entrepreneurial Activity (Number of Employees) | Length of Entrepreneurship |
|---|---|---|---|---|---|
| Ann 37 | Married 1 | Senior high (dropped out) | 17 years | Beauty and nail salon (0) | 7 years |
| Bonnie 33 | Married 1 | Senior high | 7 years | Vietnamese eatery (0) | 3 years |
| Cass 33 | Divorced 2 | Primary | 14 years | Vietnamese eatery (0) | 4 years |
| Dolly 42 | Widowed 2 | Junior high (dropped out) | 18 years | Beauty and nail salon (0) | 8 years |
| Eda 45 | Widowed 1 | Senior high | 18 years | Tailor's studio (0) | 7 years |
| Fleur 34 | Divorced 1 | Junior high (dropped out) | 14 years | Massage salon (4) | 11 years |
| Gwen 30 | Divorced 0 | Senior high (dropped out) | 6 years | Beauty and nail salon (0) | 3 years |
| Hope 35 | Married 2 | Senior high school (dropped out) | 16 years | Retail Vietnamese grocery (0) | 3 years |
| Iris 40 | Married 1 | Junior high (dropped out) | 14 years | Vegetable vendor and farmer (1) | 5 years |
| Jo 43 | Married 2 | Senior high (dropped out) | 22 years | Beauty and nail salon (0) | 13 years |
| Kyle 37 | Married 1 | Junior high | 14 years | Vietnamese eatery (0) | 6 years |
| Lori 47 | Married 2 | Senior high | 14 years | Vegetable vendor and farmer (0) | 5 years |
| Misty 42 | Divorced 2 | Junior high | 19 years | Two retail and wholesale Vietnamese groceries (3) | 12 years |

*3.2. Data Collection and Analysis*

This study is based on life story interviews, which highlighted the participants' life and work experiences in Vietnam, their motivation for moving to Taiwan, their early adaptations to employment and engagement in entrepreneurship in Taiwan, and their reflections on their personal development from creating a business. All interviews were first conducted in Chinese, recorded, and then analyzed by utilizing within- and cross-case analyses [22]. First, the researchers separately coded significant statements from each participant's interview based on the interview topic as a single case. Second, we discussed the coding and extracted the emerging categories and properties. Third, constant comparative analysis was applied to conduct the cross-case analysis to validate the emerging patterns from the within-case analysis and to identify common patterns across cases. Then, to guarantee the trustworthiness of the data analysis, the participants confirmed the appropriateness of the findings and the interpretation [23].

## 4. Results

*4.1. Establishing Micro-Entrepreneurship as a Means of Surviving and an End of Career*

For these women, who came from a self-employment culture, creating entrepreneurship was not only a means of fulfilling their expected gendered roles but also an end to pursue their individual career goals.

### 4.1.1. Coming from Self-Employment Culture

These women, who came from less economically developed areas, were familiar with self-employment because of their early entrepreneurial socialization. In their hometowns, most of their families and neighbors had earned a living by working as own-account workers and micro-business employers, for example, through running food and vegetable stands or catering shops, due to the limited wage work opportunities and the popular work ethos that, rather than working as salaried laborers, people could generate more income as entrepreneurs. Kyle noted, "My hometown is far away from the downtown in northern Vietnam, where there were few factories before I married. Some of my neighbors did fishery and agriculture farming and some ran stalls or groceries . . . My acquaintances in my home-town preferred to create their own business because they believed, by running a business, people would make more money without leaving for salaried jobs in remote towns".

Additionally, within Vietnamese culture, commercial activities are usually linked to family-based resources, including low-cost family laborers and female-gendered domestic responsibilities [19]. Thus, before migration, most participants had worked as family workers and/or established their own business to contribute financially to their family. As Misty noted, "Since my childhood, I had assisted my parents in their fruit shops. When studying in junior high, I helped my elder sister run her catering shop . . . Some years later, I learned tailoring downtown to be a good wife in the future and to run my business to support my family expenses. Later, I ran a tailor shop until I was married".

### 4.1.2. Fulfilling Expected Gendered Roles

These women's employment was partly driven by their need to contribute to the finances of their families in Vietnam and in Taiwan because, within Vietnamese culture, both unmarried and married daughters are encouraged to take financial responsibility for supporting their natal family's household [24], and their Taiwanese husbands usually could not afford family expenses alone. Additionally, being a mother/wife/daughter-in-law entails that these marriage immigrant women's sociocultural role obligations shape essential parts of their motivation as entrepreneurs. As Eda noted, "Since I was young, I had observed that my mom, as a married woman, was asked by my grandparents and father to do all the chores and care for the entire family, although she worked outside . . . Early in my immigration, my in-laws always demanded that I take all domestic responsibilities while I was busy working to afford the family expenses." In a Taiwanese patriarchal family, this is particularly true for commercially arranged marriage immigrant women because they are usually viewed as wives and mothers. Although most participants could find work easily and obtain wages similar to those of Taiwanese workers, the desire to ensure their family's wellbeing drove them to entrepreneurship because of its flexible work schedule and space. "Working as an employee in a massage salons, I had to work all night so I couldn't care for my schoolboy. Thus, I decided to run my own salon so that I could combine caring for my son and work", Fleur said.

### 4.1.3. Pursuing Career Goals with Aggressive Ambition

Immigrants are the people who are more likely to have strong motivations for seeking a better life and achieving their career goals, with additional encouragement to undertake daily risks [11]. In this way, marriage immigration in Asia could suggest a self-selection process. In this study, most participants were women with aggressive and courageous personalities, who tended to pit themselves against daily challenges and take risks. "I feel most Taiwanese are too timid to try something tough . . . I would like to try it instead of waiting for my doom. You must be bold enough to run a new business. If your business doesn't succeed, you just work as an employee again", said Misty.

For these women, engaging in entrepreneurship was not only a means of obtaining financial security but also an end to pursuing their occupational goals in their receiving country. Following emigration, these career-oriented women gradually affirmed their vocational goals of establishing their business. For Gwen, Hope, and Misty who both had

conducted their own businesses in Vietnam, entrepreneurship in Taiwan entailed fulfilling their life purpose in addition to generating income. "I strongly desired to own my store in Taiwan and even expected to expand the business. Operating a business is my career goal", Hope emphasized.

### 4.2. Negotiating Space in the Mainstream Market

These immigrant women worked as managers of their demanding micro-businesses, which targeted the mainstream market, through Taiwanese customer networks, their institutional learning, and their perseverance. In running entrepreneurships, they constantly negotiated their identities.

### 4.2.1. As the Main Managers of Micro-Businesses in Feminine Sectors, Targeting the Mainstream Market Based on Ethnic Grounds

These women's marriage migrant status reflected the limited financial, human, and cultural capital that attracted them to the micro-businesses in low-yielding sectors. They all felt capable in and suited for those sectors that are considered traditionally feminine, such as beauty, catering, and tailoring [7]. In fact, "These sectors have also attracted many Vietnamese immigrant women in America and Europe", Iris said.

Despite inspiring themselves to own a business in Taiwan, most participants did not establish their enterprises until they accumulated the human capital, funds, and social contacts necessary for starting ventures while working as salary workers. Notably, the five participants in the beauty sector initially had joint ownership with individuals from Taiwanese or other ethnic groups. "Since my partners could share the operation cost and we could cooperate in the service process . . . For each customer, I did a nail treatment while my partner gave a facial. However, the partner and I argued over managing ideas. Finally, equipped with more beauty professionalism and funding, I independently established my business", noted Ann.

These married immigrant women were responsible for creating, managing, and operating their ventures with the help of their other family. Instead of operating ethnic businesses that satisfied Vietnamese groups' needs, all of these women broke out to reach the mainstream Taiwanese market because of the small and geographically scattered Vietnamese population. "My Vietnamese catering shop isn't limited to my Vietnamese group but targets Taiwanese customers. I had been back to Vietnam to learn traditional Vietnamese cuisine, but the Taiwanese customers didn't like that strong flavor. So, my catering changed to adapt to Taiwanese customers' light tastes", said Cass.

In targeting Taiwanese demand to run their business, these female immigrant enterprisers admitted that their Vietnamese background was advantageous rather than troublesome. Most of their enterprises were built on their ethnic culture, such as Vietnamese catering, grocery, and even farming, while some performed a different specialization, such as cosmetology, which Vietnamese immigrant women commonly perform in host countries [24].

### 4.2.2. Taiwanese Customers as a Mediator to Operating Entrepreneurship

The participants expressed how their business experiences had been embedded in host society connections and ethnic networks. Although at times these women were discriminated against due to their commercially arranged marriage immigrant status, some Taiwanese individuals tended to admire Vietnamese immigrant women for their industrious working attitudes and their commitment to fulfilling family responsibilities [25]. Hence, some Taiwanese customers who empathized with these women entrepreneurs became their close friends so that the networks of the regular Taiwanese customers were essential sources of encouragement and professional information for the participants, especially for those without close family in Taiwan. As Eda, whose husband died two months after they were married, said, "I seldom contact co-ethnic friends, but some Taiwanese regulars, who help me with daily life comfort, business problem solving, and even financial support, are the benefactors in my life."

Additionally, sisterhood among Vietnamese marriage immigrant women in Taiwan helped some participants maintain their business, while the other participants seldom maintained close contact with their ethnic community because they were busy and most of their Vietnamese friends did not approve of their entrepreneurship. Hope noted, "I don't like my Vietnamese friends. They are jealous of my owning the nail shop and are usually picky about my services."

### 4.2.3. Learning to Be Desirable Entrepreneurs

In the process of working as entrepreneurs, these women learnt how to create, sustain, and develop their business in Taiwan. Female immigrant settlement services in Taiwan, such as vocational training programs and entrepreneurship guidance, also played a key role in these immigrant women's entrepreneurship experiences. These settlement services, as national apparatuses, effectively improve female immigrants' entrepreneurial capability and help to assimilate them by reproducing the Taiwanese workplace and business culture [26].

The Taiwanese government has provided government-funded training programs, especially for immigrant women, since 2005 [6]. Despite juggling work and family, over half of the participants attended vocational programs to improve the professional ability required by their business. Furthermore, their employment behaviors were reshaped by valuing Taiwanese labor market culture and abandoning aspects of their original cultures. Kyle stated, "In the cuisine training for immigrant women, the instructors usually taught us immigrant women to offer customer-oriented services, especially for Taiwanese customers. They always reminded us of the importance of food sanitation by pointing out how some cooking habits in southeastern Asia are unsanitary."

Furthermore, immigrant women were provided with minimal entrepreneurial support from the government, which suggests that the Taiwanese government might view immigrant women laborers as low-paid wage earners rather than potential entrepreneurs. It was not until 2016 that immigrant women who were nominated by female immigrant settlement agencies could garner entrepreneurial resources from the "Entrepreneurship Accelerator for Immigrant Women Program", which is funded by the government and includes entrepreneurship courses, loans, and guidance [6]. Some participants learned to negotiate entrepreneurial resources via the institutions that control these resources by displaying a mainstream self-employment culture and establishing close ties with the institutions. "To be recommended for the entrepreneurship program, I had already learned to build a good relationship with the immigrant settlement agency staff by attending more activities of the agency and providing them more assistance, like sponsoring my handmade food or working as a volunteer", said Bonnie.

### 4.2.4. Running the Business by Perseverance and Sustainable Resilience

The entrepreneurship of these women was located on the margin of the Taiwanese economic market, which is well known worldwide for its micro-businesses and small entrepreneurships [27]. In the process of business creation, sustainment, and development, these women with little capital and in a disadvantaged position faced much difficulty, such as a lack of labor and knowledge of business and finance, an unstable income, and even a husband's disproval of their entrepreneurship. "My ex-husband felt doing manicures was dirty and asked me to be wage-employed due to the fixed leave for the family. He didn't support the family expense and always stopped me from going to the salon for work", said Gwen. In addition to certain networks, it was the perseverance and resilience of these women, which was primarily rooted in their struggles against the prior hardships in Vietnam, that allowed them to support themselves in operating their businesses. Dolly, whose husband died 6 years before the study, noted, "Since childhood, I had always seen how my parents persisted in surviving by overcoming life problems . . . Despite initially feeling sad when facing many obstacles to running my salon, I quickly picked up the pieces and fought for my salon again".

Additionally, these women's perseverance and sustainable resilience were reinforced by their motherhood. Similar to other participants whose business had failed, Hope stated, "Despite having failed in business twice, I never give up because I strongly desire to offer my children a prosperous future. This expectation mitigates the suffering of running the business and my daily life."

### 4.2.5. Negotiating Voice and Identity

All of the participants initially assumed that they needed to take on the Taiwanese mainstream self-employment culture of action and value while concealing their original cultural habits because they "lacked self-confidence in working as Vietnamese immigrant women entrepreneurs and desired to integrate into the Taiwanese business market soon", as Kyle said. The participants utilized silent tolerance and humility to navigate the Taiwanese micro-business norms, which privileged Taiwanese-centric customer-oriented services. Gradually, most participants felt lost and had trouble running their entrepreneurship without their self-identity. "Always silently tolerating some Taiwanese customers' irrational behaviors made me lose my way while running my beauty business. Finally, I understood that under such a business approach, I seemed inferior to the Taiwanese. Moreover, my beauty salon can't be sustained if it is similar to all the others without my unique characters", Jo said.

As these participants increased their Taiwanese proficiency and professional capability, obtained deep knowledge regarding the atmosphere of Taiwanese micro-business, and, because Taiwanese society admired some aspects of immigrant women's aspirations to entrepreneurship in Taiwan, their self-confidence and self-identity increased. These participants gradually realized that their ignorant tolerance of customers' irrational behavior often caused their business to be neglected or even to fail. Based on their improved self-identity, most of the participants encouraged themselves to negotiate their ideas, voicing opinions to communicate with customers and integrating their own characteristics and/or Vietnamese culture into their business to provide improved services and to enhance its visibility in the Taiwanese micro-entrepreneurship market. Ann noted, "After observing some successful beauty salons run by Vietnamese and Taiwanese women, I realized they were using professional skills, distinct characteristics, and even our Vietnamese culture to make their salons stand out. Additionally, to offer better services and to make me, as a Vietnamese owner, visible, I learned to purposefully express my ideas to my customers without fear of offending them."

In turn, this process improved their homeland cultural identity and further aided them to facilitate their ethnic sisters' employment. Lori noted, "Because my vegetable business that is promoted online is good, I was interviewed by the local media, which made me proud of myself. Now, I am intentionally growing more vegetables that are popular in Vietnam and trying to sell them to Taiwanese customers. I have helped more Taiwanese customers to understand female Vietnamese immigrants and my hometown." Jo also noted, "As my salon business got better, I felt I was able and had to help my fellow ethnic sisters in Taiwan. I hired my ethnic members with the same salary as Taiwanese workers and offered some immigrant sisters beauty training at low tuition fees".

### 4.3. Obtaining Self-Empowerment While Struggling with Unexpected Challenges

Most of these immigrant women believed that self-employment empowered them to control their lives as female marriage immigrants, granting them facilitated self-identity, personal independence, social integration, family status, and ethnic sisterhood, although they still struggled with certain unexpected challenges.

Through running an entrepreneurship, these immigrant businesswomen generated more income to financially benefit their household, and some even became their family's main earners. Moreover, these businesswomen became increasingly integrated into Taiwanese society through extended social networks, which contributed to their formal financial support. Despite facing many challenges, all participants were enthusiastic about

entrepreneurship because of their passion for their business, and they were continuously self-improving their professionalism. The participants who hired employees enhanced their confidence in their own leadership and training ability. These achievements fostered the host society's respect for the women and contributed to their self-empowerment through self-confidence, self-autonomy, and economic independence.

Dolly said, "After my husband died, I had to raise two daughters myself and some friends looked down their nose at me. Through demanding self-employment, I save more money for my children's education and feel more respected by Taiwanese and Vietnamese friends". Hope noted that, "After running my salon for a few years with good business credit, I got a bank loan more easily". Like most participants, Lori confidently stated, "In the process of running the vegetable stand, I overcame many problems, like finances, planting, and struggling with my drunk and gambling husband, which made me independent and powerful".

Running an entrepreneurship also influenced these women's gendered family relations. Business achievements allowed these women to earn respect from their husbands, in-laws, and children and to improve their position in their patriarchal family. Bonnie said, "Before, my father-in-law belittled me and my child, influenced by my in-laws, and didn't follow my parenting due to my immigrant woman status. However, now, my child and father-in-law usually help in the catering shop in the evening or on the weekends".

Nevertheless, a small number of the participants who had poor marriage relationships or whose husbands originally opposed their pursuit of entrepreneurship experienced more family conflict than before, and some even ended their marriage without fearing criticism from their host community. "Actually, I had a bad marriage, and my husband hated for me to open the store. When I first ran the store, he helped me but usually opposed my business decisions, so we always argued terrifically. I opened the two grocery stores myself, and I didn't want to be controlled by him to run the business. Finally, I insisted on divorcing him because I wanted to be by myself", Misty noted.

Most participants mentioned that the business process allowed them to expand and strengthen the sisterhood among their female co-ethnic group in Taiwan. Although most of these participants' business focused on Taiwanese customers, their shops usually worked as gathering places for their female co-nationals to gather to comfort each other; meanwhile, these female co-nationals worked as flexible and low-salaried laborers to assist in the business. Cass said, "In the afternoon, when my Vietnamese catering shop is closed, my Vietnamese sisters come here with new ethnic friends to chat and sing . . . When I am busy in the shop, some Vietnamese sisters help me clean the tables and wash dishes voluntarily".

In addition, a small number of the participants who created successful businesses served as role models for other Vietnamese immigrant women, especially their co-ethnic employees, which further developed their original cultural identity. Jo, who worked as a beauty salon employer, noted, "My Vietnamese employees envied me my good business and wanted to run a salon like me. I always encouraged my skillful Vietnamese employees to create their own salons and helped them obtain bank loans, improve their professionalism, and introduce them to beauty products, but only if they didn't open their salons nearby. I felt proud that my ex-employees started their salons and that we, as Vietnamese immigrant women, could have a successful career like Taiwanese women".

Notably, despite being satisfied with their entrepreneurship, running a business entails full-time involvement, economic risks, and certain issues of managing employees that could conflict with the original expectations of these businesswomen. Hence, they had to struggle to juggle work, family, and self-health while seeking more financial resources and managing the human resources of their shops. "Unexpectedly, I must work over 12 h, daily, from Monday to Sunday to maintain my business. I feel guilty for not staying long with my children, and my health declines due to demanding work . . . I also have to deal with my employees' turnover and incompetence, which damage the business", Misty noted.

## 5. Discussion

### 5.1. Vietnamsese Immigrant Women Being Pushed and Pulled toward Self-Employment

Most research that has investigated male immigrants' motivations for creating business has typically highlighted the employment macro-context in host societies and the resources and career ambitions that immigrants possess [28]. Although in this study some of the immigrant women's business motives were similar to those of male immigrants, attempting to balance family responsibility and work was salient in female immigrants' discourse [8], driven by their traditional gendered role expectations in familial and social contexts. In this respect, these immigrant women seemed pushed toward entrepreneurship. However, that these female immigrants' pursuit of business was also enforced by their career self-fulfillment suggests that entrepreneurship may function as an opportunity strategy for them. Hence, these women were both pushed and pulled toward self-employment by not only macro-contexts and personal factors but also by the values of their home and host societies [29]. Therefore, there is no single theory that can comprehensively explain these women's complicated motivations for entrepreneurship. However, the result of this present study is different to Billore's [30] finding that well-educated Indian immigrant women move to Japan under family reunification with their Indian husband. These Indian immigrant women in Japan tend to be pulled towards entrepreneurship because of serving their ethnic community, and their Indian husbands encourage and support this.

### 5.2. Vietnamsese Immigrant Women's Business Roles Contradicting the Gendered Roles

In this study, the immigrant women, who were originally expected to be compliant housewives, were the primary managers of a business with the help of other family members. However, in other host societies, immigrant women migrating under family reunification have traditionally worked as subordinate resources, available to their husbands in a family entrepreneurship [31]. One possible explanation for this result may be that most of these women's native spouses were economically disadvantaged so that they neither possessed relevant social contacts or personal/professional capital, nor were they encouraged to launch their own ventures nor acquired beneficial resources to support their wives' businesses [25]. Hence, these women had to take essential responsibility for running their entrepreneurship by themselves. The solid business roles that these immigrant women played were consistent with their homeland's culture; in Vietnam, women were encouraged to be self-employed, with men's help secondary [24]. However, these women's business roles contradicted the sociocultural roles they were expected to play as married immigrant women in Taiwan. Therefore, in the initial stage of creating a business, they experienced more conflicts with their husbands and in-laws regarding housework and business until, finally, their family members abandoned attempting to dominate their business due to their lack of essential capital to run a business and these women's perseverance in their entrepreneurship. For those with poor marriage relationships, when their husbands insisted on intervening in their entrepreneurships, some women ended their marriages.

### 5.3. Native Spouses Playing a Determining Role in Vietnamsese Immigrant Womens' Business

Clearly, in the process of these women running their businesses, they and their husbands constantly negotiated their self-identities and positions amid the shifting entrepreneurial and familial contexts. Additionally, this result is inconsistent with Munkejord's [24] conclusion that Russian immigrant women marry Norwegian men of high economic-social status, who provide important resources for their wives' ventures as mediators to the host society, out of love. This suggests that although the immigrant women in this study and in Munkejord's [32] study all migrated by marrying native men in host countries, the two groups are different in their marriage backgrounds, which may result in their different life contexts. In these contexts, the micro-family context, especially regarding native spouses, plays a determining role in these married immigrant businesswomen's access to business resources [28].

### 5.4. Self-Employed Vietnamsese Immigrant Women Being Open to Host Culture

The study found that these female Vietnamese immigrant entrepreneurs, with limited capital, drew significantly on the native customers of their host society for essential support in running their businesses, which is inconsistent with some previous studies, indicating that the fewer socioeconomic resources immigrant women entrepreneurs have, the more likely they are to rely mainly on co-ethnic networks [28]. This may be explained by the fact that, to target the mainstream market, the immigrant women in this study tended to be open to their host culture and adopted a multicultural strategy that is heavily dependent on host society ties and co-ethnic support [33]. Moreover, another explanation may be that the positive and energetic images that most Vietnamese marriage women in Taiwan demonstrate cause some Taiwanese customers to appreciate and empathize with these disadvantaged women because, in some respects, these married immigrants seem to save certain socioeconomically disadvantaged Taiwanese males and families [26]. In this way, these women's disadvantaged immigrant status may be turned into advantages as they create a business. Thus, this finding suggests that ethnic background, immigrant status, gender, and class may intertwine to condition the formation of social capital and its implication for immigrant self-employment [34].

### 5.5. Vietnamese Female Immigrant Entrepreneurs' Professional Identity Being Conditioned by Institutional Recognition

This study accentuates how, in a host society, female immigrant entrepreneurs' professional identity and government resources are conditioned by institutional recognition that significantly influences their self-employment opportunities and positions. These women's formal and informal learning was dominated by institutional processes that favor their host culture. However, immigrant women may be viewed as being culturally deficient, whereby desirable learning for them is defined as consistent with the host and institutional culture [35]. Namely, immigrant businesswomen's learning, which was enforced by their pursuit of running an entrepreneurship, may become a kind of symbolic violence, functioning as a mechanism to legitimate the existing social order in their host self-employment market [36].

### 5.6. Self-Employment Not Being a Taken-for-Granted Choice for Vietnamese Immigrant Women to Balance Family and Work

The immigrant women in this study originally planned to spend more time caring for family by running a business. However, most of them decided to work long hours to maintain ventures while negotiating the tensions of running a business and fulfilling domestic responsibilities, despite facing family conflicts and their own guilt. In addition to their total investment, this finding underscores that these businesswomen persevered in maintaining entrepreneurship due to their strong career ambition and work commitment. Furthermore, previous studies also have found that immigrant businesswomen had to work excessive hours to balance work and domestic duties [11]. Clearly, despite the indisputable allure of entrepreneurship, it may not be a taken-for-granted choice for female immigrants who seek to balance family and work.

### 5.7. Business Immigrant Women's Sustainable Agency Being Emphasized

Most studies have emphasized immigrant women's vulnerability to the unfairness and segregation of host labor markets due to the essential influences of contexts and gender on their businesses [28]. Nevertheless, in this study, despite being vulnerable to discrimination, interwoven with their gender, nationality, and marriage immigrant status in their host country, these female Vietnamese immigrant entrepreneurs exercised their sustainable agency to fulfill different life meanings for themselves and their families, rather than being submissive to the host culture by continuously negotiating their identities and navigating their positions while running a business. By creating their own entrepreneurship, from which they derived high job satisfaction, these women were empowered with self-

autonomy, an improved self-identity, and an enriched homeland cultural identity, although some faced marriage crises and business challenges. Therefore, these results extend the feminist perspective on entrepreneurship with a focus on the sustainable agency and sense of competence of female immigrants in realizing their career goals through the pursuit of self-employment in their receiving countries.

## 6. Conclusions and Implications

In this study, female Vietnamese marriage immigrants in Taiwan were pulled and pushed to create demanding micro-entrepreneurships through their self-employed socialization to fulfill multiple family obligations and achieve career goals. Targeting their host market, these women managed their businesses according to Taiwanese customer networks and institutionalized learning, their sustainable agency, and their shifting self-identity. Managing an entrepreneurship is an avenue for empowering these women, favoring self-identity, social integration, and position within the gendered boundaries of family expectation while they had to combat unforeseen challenges. These businesswomen, who escaped from the stereotypes of obedience and marginalization [7], may contribute to the positive image of female marriage immigrants in Taiwan. Clearly, the individuals and certain significant others played essential roles in these marriage immigrant women's entrepreneurial experiences, which were also profoundly shaped by their homeland culture and prior socialization, their life experiences and positions in Taiwan, and their shifting social relationships as they gradually engaged in business practices by negotiating their self-identities and sense of meaning.

The optimal institutional settlement service for immigrant women is to enhance their acculturation instead of assimilation in a host country [26]. Therefore, supporting immigrant women with multicultural settlement services at the ground level is essential, specifically, offering business training that appreciates and integrates immigrants' cultures to create platforms where immigrant businesswomen exchange entrepreneurial experiences and opportunities. Additionally, the indistinct futures of and the failures in maintaining businesses among marriage immigrant businesswomen warrant further investigation.

**Funding:** This research was funded by the Ministry of Science and Technology, grant number MOST106-2511-S-020-001.

**Institutional Review Board Statement:** Ethical review and approval were waived for this study because this study is a non-interventional one. However, all the participants in this study were fully informed that their anonymity and privacy were assured, why the research was conducted and how their data would be used. Besides, informed consent was obtained from all the participants.

**Informed Consent Statement:** Informed consent was obtained from all subjects involved in the study.

**Data Availability Statement:** The data are not publicly available due to the privacy of the participants.

**Acknowledgments:** I deeply appreciate the Vietnamese immigrant women's participation in the research.

**Conflicts of Interest:** The authors declare no conflict of interest.

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
