# Peer review of "Entrepreneurship Experiences among Vietnamese Marriage Immigrant Women in Taiwan"

_sustainability, doi:10.3390/su14031489_

Round 1

Reviewer 1 Report

This paper looks into entrepreneurship experiences among Vietnamese marriage immigrant women in Taiwan. Vietnamese marriage immigrant women are found in several countries in the region such as South Korea, China, and Taiwan. We are familiar with other aspects of these migrants such as migration process, experiences, and integration issues. However, we are not familiar with the economic incorporation of these immigrants, especially the entrepreneurship dimension. This study is thus an innovative study that addresses the gap in the existing literature. This paper is well-structured supported by sufficient theoretical insights and empirical findings. The qualitative data is rich and reveals the whole experiences of these immigrants' entrepreneurship process. Having said that, I am sharing two  of concerns that the author must consider when submitting the final manuscript: 

1. We are not familiar with Vietnamese migration to Taiwan. A section on Vietnamese migration to Taiwan will help us contextualize the experiences of these immigrants in a nuanced way. 

2. It is true that immigrant entrepreneurship is not thoroughly studied in Asia-Asia migration. However, there exists some literature on immigrant entrepreneurship in East Asia in the context of male migration such as  South Korea and Japan. The author may like to consult such literature and compare and contrast the findings with Vietnamese women entrepreneurs. Currently, it seems that there is no such study in Asia as the author pointed out. 

Reviewer 2 Report

This is an interesting and compelling study. Identified a gap in the research. Some of the word use and sequential words are a little confusing. On pg. 2 line 51, you use “family reunion” instead of family reunification. I’m not sure if there is a reason, but I haven’t seen that before. It also appears again in discussion on pg. 12. Also, on pg. 2 when describing past research, there are words typically used for listing things “next” on pg. 59. I also haven’t seen this done unless it’s describing a linear process. The purpose and research questions are clear.

For sociocultural theory, a source you may find interesting is: Esmonde, I., & Booker, A. N. (Eds.). (2016). Power and privilege in the learning sciences: Critical and sociocultural theories of learning. Routledge. https://doi.org/10.4324/9781315685762 This source discusses using sociocultural theories with critical theories, similar to what you are doing.

Methodology: Add a citation to the first sentence “A qualitative interpretive approach (add citation) was employed…” Add citation to sampling method. I appreciate the participant table.

Results: For results, it seems that 4.1 needs a developed paragraph before moving into 4.1.1. There are interesting quotes and findings.

Discussion line 416, “marriage” may not be the intended word. The discussion would benefit from subheadings to help the reader understand and keep content organized. Organizing the discussion using the research questions as headings would be helpful.

Conclusion and Implications: what about implications for future research?

Reviewer 3 Report

Review report

The paper addresses a topic of interest, namely the field of immigrant female entrepreneurship, the chosen subject being the entrepreneurial experiences among Vietnamese married immigrant women from Taiwan.

The paper has a number of strengths, namely

  1. Clear and reasoned presentation of the study topic and research questions.
  2. The qualitative research undertaken; the primary data collected being well analysed.
  3. The Discussion section of the paper, which is clear, reasoned and in accordance with relevant references.
  4. Current and relevant references.

My suggestions are as follows

  • The Study Framework section needs to be further developed with specialized bibliography in the field of immigrant entrepreneurship. This topic is missing in the current approach.
